# CAT: Contrastive Adversarial Training for Evaluating the Robustness of Protective Perturbations in Latent Diffusion Models

**Sen Peng** [1]  **Mingyue Wang** [1]  **Jianfei He** [1]  **Jijia Yang** [1]  **Xiaohua Jia** [1]

## Abstract

Latent diffusion models have recently demonstrated superior capabilities in many downstream image synthesis tasks. However, customization of latent diffusion models using unauthorized data can severely compromise the privacy and intellectual property rights of data owners. Adversarial examples as protective perturbations have been developed to defend against unauthorized data usage by introducing imperceptible noise to customization samples, preventing diffusion models from effectively learning them. In this paper, we first reveal that the primary reason adversarial examples are effective as protective perturbations in latent diffusion models is the distortion of their latent representations, as demonstrated through qualitative and quantitative experiments. We then propose the Contrastive Adversarial Training (CAT) utilizing lightweight adapters as an adaptive attack against these protection methods, highlighting their lack of robustness. Extensive experiments demonstrate that our CAT method significantly reduces the effectiveness of protective perturbations in customization, urging the community to reconsider and improve the robustness of existing protective perturbations. The code is available at `https://github.com/senp98/CAT`.

## 1. Introduction

Diffusion-based image generation models, especially the Latent Diffusion Models (LDMs) (Rombach et al., 2022), have exhibited strong capabilities across various downstream image generation tasks, including text-to-image synthesis (Dhariwal & Nichol, 2021; Rombach et al., 2022), image-to-image translation (Saharia et al., 2022; Zhang et al., 2023), image editing (Kawar et al., 2023), and image inpainting (Lugmayr et al., 2022). By utilizing the open-source LDM such as Stable Diffusion (Stability.ai, 2024) as the pre-trained model, users can customize image generation models on their datasets at a significantly lower cost than training from scratch. Despite their prominence and popularity, the customization of LDMs also raises significant concerns regarding privacy and intellectual property rights (Dixit, 2023; LLP, 2023). It can be misused to generate fake human face images of a specific identity (Li et al., 2022) or mimic the style of a specific artist by utilizing unauthorized data. Undoubtedly, such malicious customization severely compromises the rights of data owners and raises ethical concerns across the community.

Several protective perturbation methods (Salman et al., 2023; Liang et al., 2023; Shan et al., 2023; Le et al., 2023; Liang & Wu, 2023; Xue et al., 2024; Liu et al., 2024) have been proposed to address these concerns. These methods introduce imperceptible perturbations to image samples in the customization dataset, creating adversarial examples that effectively degrade the performance of downstream customization tasks. Although significant progress has been made in developing perturbation methods, the robustness of these protection techniques has received limited attention. Recent studies (Zhao et al., 2024; Cao et al., 2024; Hönig et al., 2025) have investigated the resilience of protective perturbations against purification-based adaptive attacks. By purifying the protected images before customization, these methods effectively compromise the robustness of existing protective perturbation. However, existing purification-based adaptive attacks depend on optimization, resulting in high computational costs. Without an efficient mechanism to detect the protected perturbed images, purification must be applied to every sample in the customization dataset, increasing resource demand. Furthermore, purification without prior knowledge of the original data distribution introduces uncertainty, potentially altering the learned distribution. In extreme cases, optimization-based purification may entirely replace the original image content if the substituted one achieves a lower loss for the optimization objective.

In this paper, we introduce Contrastive Adversarial Training (CAT) from a novel perspective, focusing on model adaptation rather than purification as an adaptive attack. We

---

[1]Department of Computer Science, City University of Hong Kong, Kowloon, Hong Kong SAR. Correspondence to: Sen Peng <senpeng2-c@my.cityu.edu.hk>.

*Proceedings of the 42nd International Conference on Machine Learning*, Vancouver, Canada. PMLR 267, 2025. Copyright 2025 by the author(s).

first reveal that the primary reason adversarial examples are effective as protective perturbations in LDM customization is that their latent representations are distorted. Building on this insight, we develop the CAT method targeting the latent autoencoder of the LDM, utilizing the contrastive adversarial loss to minimize the reconstruction loss of protected image samples. During adversarial training, we introduce LoRA adapters (Hu et al., 2022) to specific layers of the latent autoencoder, ensuring that only the adapter weights are modified. This approach expands the parameter space to mitigate the effectiveness of adversarial examples and reduces computational costs, enhancing its adaptability across different scenarios. Extensive experiments demonstrate that applying CAT decreases the effectiveness of protective perturbations, urging the community to reconsider their robustness. In summary, our main contributions are as follows:

1. We identify that the primary reason adversarial examples effectively serve as protective perturbations in LDM customization is the distortion of their latent representations.

2. We propose contrastive adversarial training with adapters as an adaptive attack to evaluate the robustness of existing protective perturbation methods from a novel perspective of model adaptation rather than purification.

3. We conduct extensive experiments demonstrating that applying CAT can significantly reduce the effectiveness of existing perturbation methods, urging the community to reconsider their robustness.

## 2. Related Works

### 2.1. Diffusion-based Text-to-Image Models

Diffusion-based text-to-image models have demonstrated remarkable capabilities across various downstream tasks. They learn to generate images by simulating a forward diffusion process that adds noise, and a reverse diffusion process that removes it, typically using a U-Net (Ronneberger et al., 2015) as the backbone (Ho et al., 2020). Latent diffusion models (Rombach et al., 2022) improve training and inference efficiency while preserving generation quality by encoding samples into their latent representations via an autoencoder. Stable Diffusion (Stability.ai, 2024) was later released as an open-source LDM, which significantly boosted the development of various downstream applications in diffusion-based text-to-image generation.

### 2.2. Customization of LDMs

LDMs can be customized for a range of downstream tasks, which broadly fall into text-driven image synthesis and editing. Text-driven image synthesis is typically categorized into object-driven image synthesis and style mimicry (Peng et al., 2024). The former focuses on generating images containing specific objects described by the text prompt,

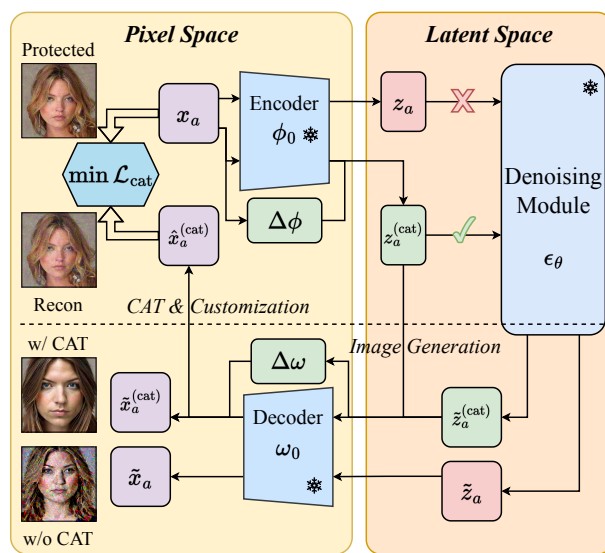

*Figure 1.* Overview of the proposed CAT method. The CAT adapters ($\Delta\phi$ and $\Delta\omega$) are integrated into the latent encoder and decoder to realign the latent representations of protected samples. Instead of directly using $z_a$ for the denoising module in the diffusion process, the protected image $x_a$ is first encoded into the realigned latent representation $z_a^{(\text{cat})}$ through the adapted encoder, then decoded into $\hat{x}_a^{(\text{cat})}$ using the adapted decoder. This process ensures that the generated image $\tilde{x}_a^{(\text{cat})}$ is correctly decoded into the pixel space, effectively neutralizing the protective perturbations.

while the latter aims to generate images that reflect the style specified in the text prompt. DreamBooth (Ruiz et al., 2023) customizes text-to-image diffusion models by fine-tuning both the U-Net and text encoder using a few target subject images. It introduces a class-specific prior preservation loss to prevent overfitting on the subject, enabling the model to generate diverse class images while preserving key visual features of the subject. LoRA (Hu et al., 2022), originally developed to enable parameter-efficient fine-tuning large language models, has been adapted to diffusion models to enable customization via low-rank adaptation. By introducing lightweight trainable adaptation layers into a frozen pre-trained model and updating only these layers, LoRA significantly reduces the memory and computational overhead during model customization.

### 2.3. Protective Perturbations

Protective perturbations build upon the idea of adversarial attacks (Goodfellow et al., 2015) in classification models, which introduce imperceptible input perturbations to mislead models into incorrect predictions. Recent studies (Salman et al., 2023; Liang et al., 2023; Shan et al., 2023; Le et al., 2023; Liang & Wu, 2023; Xue et al., 2024)

have shown that diffusion-based generative models are also vulnerable to such attacks. In this context, adversarial examples serve as protective perturbations that significantly degrade the performance of downstream tasks and prevent unauthorized customization. Photoguard (Salman et al., 2023) first introduces protective perturbations to increase the cost of diffusion-based image editing methods such as SDEdit (Meng et al., 2022). The perturbation is generated by jointly maximizing the latent reconstruction loss and the diffusion denoising loss with respect to the protected images. AdvDM (Liang et al., 2023) and SDS (Xue et al., 2024) generate perturbations that target the entire LDM through end-to-end optimization to prevent unauthorized model customization. In contrast, Glaze (Shan et al., 2023) focuses on attacking the latent encoder of LDMs to distort the encoded representations. Mist (Liang & Wu, 2023) optimizes the joint latent and denoising losses to generate perturbations, while Anti-DreamBooth (Le et al., 2023) specifically targets the DreamBooth (Ruiz et al., 2023) customization approach. MetaCloak (Liu et al., 2024) leverages a meta-learning framework with transformation sampling to generate transferable and robust perturbations.

Purification-based adaptive attacks have recently been proposed against existing protective perturbations. IM-PRESS (Cao et al., 2024) introduces a purification technique that minimizes the difference between the protected image and its reconstruction from the latent autoencoder. GrIDPure (Zhao et al., 2024) extends DiffPure (Nie et al., 2022) to purify protected samples before customization, while Noisy-Upscaling (Hönig et al., 2025) achieves a similar goal using other generative models. However, all these methods decrease the effectiveness of protective perturbations through purification-based adaptive attacks. To the best of our knowledge, this is the first work to investigate the robustness of such perturbations in the context of LDM customization from the novel perspective of model adaptation. Recent RobustCLIP (Schlarmann et al., 2024) enhances the robustness of CLIP-based vision encoders through adversarial training, particularly against perturbations that disrupt text-image semantic alignment. While effective, it requires fine-tuning the entire encoder on a large set of adversarial examples to achieve general robustness. In contrast, CAT introduces lightweight adapters during customization on protected data. These adapters are detachable at inference time, leaving the original model performance unaffected.

## 3. Understanding Protective Perturbations

### 3.1. Threat Model

We first introduce the threat model considered in this work. We assume the data owner holds clean data samples and intends to share them publicly. To prevent unauthorized customization of the generative model, the owner applies a protective perturbation to each image before release and then publicly shares the perturbed dataset. In this setting, the adversary only has access to the protected data and is unaware of the specific protection method used by the data owner. We want to demonstrate that, even without knowledge of the protection technique and with access only to the protected data, the adversary can still successfully customize the LDM using our proposed CAT.

### 3.2. Distortion in Latent Representations

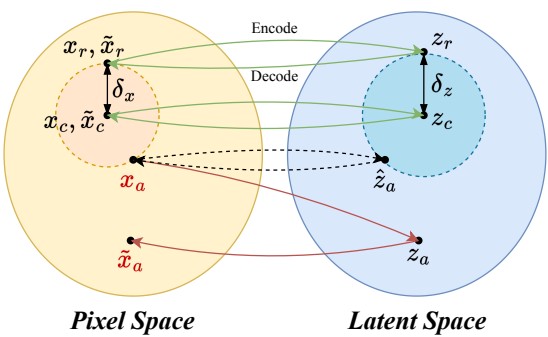

*Figure 2.* Distortion of latent representations for protected image samples in LDMs. Given a clean image $x_c$, it is encoded into a latent representation $z_c$ and decoded back to pixel space as $\tilde{x}_c$. For the protected image $x_a$, the latent representation $\hat{z}_a$ is distorted, leading to a reconstructed image $\tilde{x}_a$ that deviates significantly from $x_a$, resulting in a larger reconstruction error.

We first identify a key reason why adversarial examples serve as effective protective perturbations in LDMs: their representations are distorted in the latent space. As illustrated in Figure 2, the latent autoencoder of a LDM first encodes a given clean image $x_c$ from pixel space into a latent representation $z_c$. Subsequently, the denoising module takes the latent representation as input because it has significantly lower dimensionality while still preserving the semantic features. Additionally, it requires that the autoencoder can correctly decode the latent representation $z_c$ back to pixel space, ensuring that the reconstructed image $\tilde{x}_c$ matches the original image $x_c$ in the ideal case. However, the mappings between pixel and latent space are distorted for adversarial examples. For an adversarial example $x_a$ within a perturbation budget $\delta_x$, the latent representation $\hat{z}_a$ should remain close to $z_c$ within a distance $\delta_z$ and correctly decode back to $x_a$ in pixel space in the ideal case. However, $x_a$ is instead mapped to a distorted latent representation $z_a$. This distorted representation is decoded to $\tilde{x}_a$, which exhibits a significantly larger reconstruction error.

Similar observations about the latent representation are also reported in previous works (Xue et al., 2024; Cao et al.,

2024). In contrast, our study conducts more extensive qualitative and quantitative experiments to thoroughly validate the distortion in latent representations of protected samples. To illustrate this, we visualize the latent representations of various protected samples using t-SNE (der Maaten & Hinton, 2008), as shown in Figure 3. We also include the UMAP visualization (McInnes et al., 2018) in Figure 8, which is provided in Appendix A, to enhance the relative distance demonstration. We use four clean images of an identity from

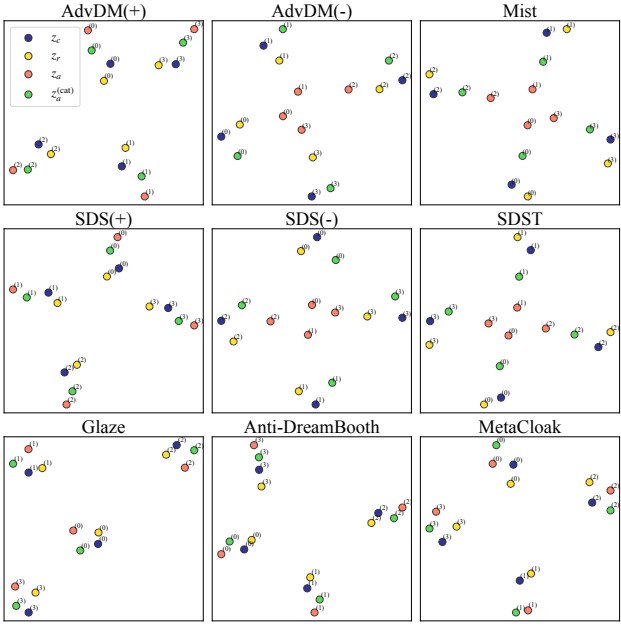

*Figure 3.* Visualization of latent representations using t-SNE of clean sample $z_c$, noisy sample $z_r$, protected sample $z_a$, and the latent representation of protected sample $z_a^{(\text{cat})}$ encoded by the latent encoder with CAT adapter. Protected samples are generated using nine protective perturbation methods, with four clean images from the VGGFace2 dataset as the base. The perturbation budget for protected and noisy samples is fixed at $\delta_x = 16/255$. Each number labels the representations obtained from the same image.

the VGGFace2 (Cao et al., 2018) dataset as an illustration. For each clean sample $x_c$, we generate its adversarial example $x_a$ as the protected image using nine different protective perturbation methods, within the perturbation budget fixed at $\delta_x = 16/255$. The protected sample is then encoded into the latent space through the encoder as $z_a = \mathcal{E}(x_a)$, while the clean sample is encoded as $z_c = \mathcal{E}(x_c)$. Additionally, we construct a noisy sample $x_r = x_c + clip(r, \delta_x)$ by adding Gaussian noise $r \sim N(0, \delta_x^2)$, clipped using the same perturbation budget $\delta_x$. This noisy sample is encoded as $z_r = \mathcal{E}(x_r)$. Finally, the adversarial sample $x_a$ is also encoded using the latent encoder with the CAT adapter (in the CAT-both setting), resulting in $z_a^{(\text{cat})} = \mathcal{E}^{(\text{cat})}(x_a)$. For each protection method shown in the subplots, $z_a$, $z_r$ and

$z_a^{(\text{cat})}$ are clustered with their corresponding clean sample $z_c$. Since four images are evaluated for each protection method, their latent representations naturally form four distinct groups. Within each group, it can be observed that the latent representation of the protected sample $z_a$ exhibits a larger distance from the clean sample's latent representation $z_c$ compared to the noisy sample $z_r$ under the same perturbation budget. This pattern holds across all four image samples for every protective perturbation method, indicating that the latent representations of adversarial examples as protected samples are distorted across all evaluated protection methods. This observation is also supported by the UMAP visualization.

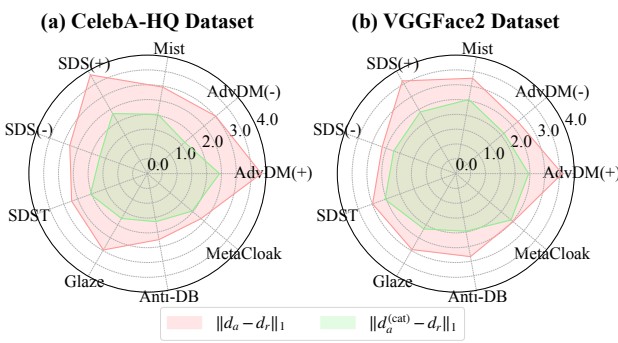

*Figure 4.* Comparison of latent representation distances $\|d_a - d_r\|$ and $\|d_a^{(\text{cat})} - d_r\|$ for various protective perturbation methods across (a) CelebA-HQ and (b) VGGFace2 datasets. The mean distance values over four images for each method are presented, highlighting the CAT adapter's effectiveness in reducing latent representation distortion.

In addition to qualitative experiments, we conduct a quantitative analysis of latent representation distances to support our findings. We define the MAE distance between the latent representation of a sample and its corresponding clean sample as $d$. Specifically, the distance between $z_a$ and its corresponding $z_c$ is denoted as $d_a = \|z_a - z_c\|_1$, while the distance between $z_a^{(\text{cat})}$ and its corresponding $z_c$ is represented as $d_a^{(\text{cat})} = \|z_a^{(\text{cat})} - z_c\|_1$. The distance between a noisy sample $z_r$ and its corresponding $z_c$ is denoted as $d_r = \|z_r - z_c\|_1$. For comparison, we present the relative differences between $d_a$ and $d_a^{(\text{cat})}$ over $d_r$, as shown in Figure 4. We observe that $d_a$ is consistently larger than $d_r$ across all evaluated protection methods, indicating that adversarial examples significantly distort their latent representations compared to noisy samples with the same perturbation budget. Furthermore, our proposed CAT adapter effectively reduces this distortion, realigning the latent representations of protected examples toward their original positions.

### 3.3. Learnability of Distorted Latent Representations

We have identified that a key reason for the effectiveness of adversarial examples as protective perturbations is the distortion of their latent representations. Building on this finding, we seek to explore a further question: Can the latent representations of adversarial examples be effectively learned by the diffusion process?

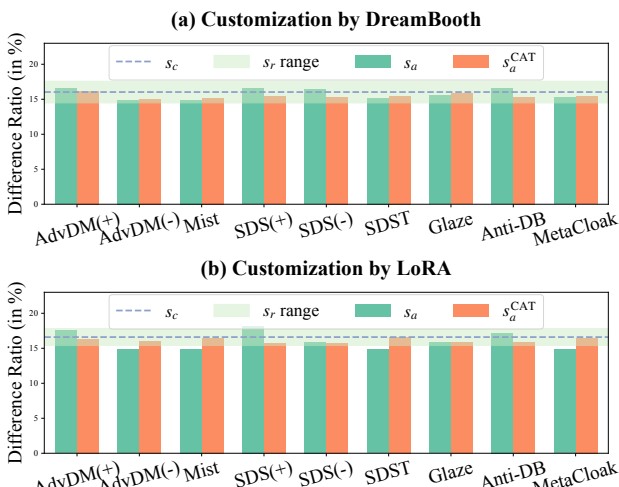

Figure 5. Difference ratios for nine protection methods, evaluated using an image from the VGGFace2 dataset. Results are shown for two fine-tuning approaches: (a) by DreamBooth and (b) by LoRA. The $s_r$ range $[s_c - \|s_c - s_r\|, s_c + \|s_c - s_r\|]$ is shaded, demonstrating that $s_a$ consistently falls within or near this range across all methods, indicating that both clean samples and adversarial examples can be effectively learned by the diffusion process.

To explore this, we design the following experiment: diffusion models are known to overfit when trained or fine-tuned on a very small number of image (Carlini et al., 2023). In such cases, the overfitted model exhibits memorization, generating samples that are nearly identical to the training image when given the same text prompt used during fine-tuning. We first fine-tune the pre-trained LDM on a single image obtained from one specific identity in the dataset. Once the model overfits, we generate an image from the fine-tuned model using the exact same text prompt. For the latent representation $z_c$ of a clean image, we expect the generated $\tilde{z}_c$ to be close to $z_c$ in the latent space. The difference ratio for this clean image is defined as $s_c = \|z_c - \tilde{z}_c\|/z_b$, where $z_b = \max(z_c) - \min(z_c)$ serves as a normalization term. Similarly, the difference ratios for $s_r$, $s_a$ and $s_a^{(cat)}$ are defined as $\|z_r - \tilde{z}_c\|/z_b$, $\|z_a - \tilde{z}_c\|/z_b$ and $\|z_a^{cat} - \tilde{z}_c\|/z_b$, respectively. For simplicity, we also define the $s_r$ range as $[s_c - \|s_c - s_r\|, s_c + \|s_c - s_r\|]$. We present the evaluation results in Figure 5. The difference ratios are evaluated for nine protection methods based on the sample clean im-

age from an identity in the VGGFace2 dataset and two fine-tuning approaches, DreamBooth and LoRA. As shown in Figure 5, $s_a$ consistently falls within or near the range of $s_c \pm s_r$ across all protection methods and both fine-tuning approaches, which suggests that the latent representations of both clean and adversarial samples can be effectively learned by the diffusion process. This indicates that the primary reason for the effectiveness of adversarial examples as protective perturbation methods lies in the distortion of their latent representations, rather than the limitation of the diffusion process in learning these distorted representations.

In summary, this section provides a deeper investigation of why adversarial examples of LDMs are effective as protective perturbations against unauthorized data usage. We identify that the primary reason lies in the distortion of latent representations for these adversarial examples. Furthermore, we verify that these distorted latent representations can still be effectively learned by the diffusion process, confirming that the distortion itself is the key factor behind their effectiveness. We emphasize that this conclusion is mainly based on empirical experimental observations, and other potential contributing factors may exist.

## 4. CAT: Contrastive Adversarial Training

### 4.1. Contrastive Adversarial Loss

We have observed that adversarial examples are effective as protective perturbations because their latent representations are distorted. This indicates that the latent autoencoder $\{\mathcal{E}_\phi, \mathcal{D}_\omega\}$ of the LDM is attacked. To mitigate this, we apply contrastive adversarial training to the latent autoencoder to realign the latent representations of adversarial examples. We first introduce the contrastive adversarial loss used for our adversarial training strategy as follows. Given the protected image $x_a$ and the latent autoencoder $\{\mathcal{E}_\phi, \mathcal{D}_\omega\}$, we define the loss function as

$$\mathcal{L}_{cat}(\mathcal{E}_\phi, \mathcal{D}_\omega, x_a) = \mathbb{E}_{x_a \sim p(x_a)} \|\mathcal{D}_\omega(\mathcal{E}_\phi(x_a)) - x_a\|_2^2. \quad (1)$$

The protected image is first encoded into the latent space by the encoder $\mathcal{E}_\phi$ and then decoded back into the pixel space by $\mathcal{D}_\omega$. Since the decoded image should ideally match the original protected image, we define the contrastive loss as the MSE distance between them over all given $x_a$. This loss encourages the autoencoder to correctly reconstruct $x_a$, restoring its latent representation.

### 4.2. CAT Adapters

Fine-tuning all layers of the latent autoencoder may affect the model's performance in encoding and decoding unprotected image samples. Moreover, we aim to introduce additional parameters to expand the parameter space, improving the model's resilience against adversarial examples after the

contrastive adversarial training. Therefore, we incorporate adapters in our proposed contrastive adversarial training. Specifically, we apply the LoRA (Hu et al., 2022) adapter to all convolutional and attention layers within the latent autoencoder. For the model weights $\phi_0 \in \mathbb{R}^{d \times k}$ of the latent encoder, the latent representation $z_a^{(\text{cat})}$ of a protected sample $x_a$ is given by

$$z_a^{(\text{cat})} = \phi_0 x_a + \Delta\phi x_a = \phi_0 x_a + B^\phi A^\phi x_a, \quad (2)$$

where $B^\phi \in \mathbb{R}^{d \times r}$ and $A^\phi \in \mathbb{R}^{r \times k}$ are the weights of the CAT adapter attached to the latent encoder.

Similarly, for the model weights $\omega_0 \in \mathbb{R}^{k \times d}$ of the latent decoder, the reconstruction $\hat{x}_a^{(\text{cat})}$ in pixel space of the latent representation $z_a^{(\text{cat})}$ is given by

$$\begin{aligned} \hat{x}_a^{(\text{cat})} &= \omega_0 \, z_a^{(\text{cat})} + \Delta\omega \, z_a^{(\text{cat})} \\ &= \omega_0 \, z_a^{(\text{cat})} + B^\omega \, A^\omega \, z_a^{(\text{cat})}, \end{aligned} \quad (3)$$

where $B^\omega \in \mathbb{R}^{k \times r}$ and $A^\omega \in \mathbb{R}^{r \times d}$ are the weights of the CAT adapter attached to the latent decoder. The overall design of our proposed CAT method is illustrated in Figure 1. For a protected image sample $x_a$, we do not directly use its encoded latent representation $z_a$ for the denoising module to learn in the diffusion process. Instead, we first apply CAT by encoding $x_a$ using the latent encoder attached with the CAT adapter $\Delta\phi$, obtaining the realigned latent representation $z_a^{(\text{cat})}$. This realigned representation is then decoded back using the latent decoder, which is also attached with the CAT adapter $\Delta\omega$, resulting in the reconstructed image $\hat{x}_a^{(\text{cat})}$ in the pixel space. Our proposed CAT method is optimized by solving the following objective:

$$\Delta\phi, \Delta\omega = \arg_{\Delta\phi, \Delta\omega} \min \mathcal{L}_{\text{cat}}(\mathcal{E}_\phi^{\text{cat}}, \mathcal{D}_\omega^{\text{cat}}, x_a), \quad (4)$$

where the weights for model $\mathcal{E}_\phi^{\text{cat}}$ are $\phi_0 + \Delta\phi$ and the weights for model $\mathcal{D}_\omega^{\text{cat}}$ are $\omega_0 + \Delta\omega$.

Once CAT is applied and the CAT adapters are obtained, customization on protected image samples is performed using the latent autoencoder augmented with the CAT adapters. For image generation, since the denoising module is fine-tuned on the realigned latent representation $z_a^{(\text{cat})}$, the generated output in the latent space for the given text prompt is also realigned. Consequently, it is decoded into the image $\tilde{x}_a^{(\text{cat})}$ in the pixel space instead of $\tilde{x}_a$, effectively neutralizing the protective perturbations.

## 5. Experiments

### 5.1. Experimental Setup

**Evaluated Protective Perturbation Methods** To evaluate the effectiveness of our proposed CAT method in compromising the robustness of protective perturbations, we test

CAT across nine protective perturbation settings: (1) AdvDM(+) (Liang et al., 2023), (2) AdvDM(-) (Xue et al., 2024), (3) Mist (Liang & Wu, 2023), (4) SDS(+) (Xue et al., 2024), (5) SDS(-) (Xue et al., 2024), (6) SDST (Xue et al., 2024), (7) Glaze (Shan et al., 2023), (8) Anti-DreamBooth (Le et al., 2023), and (9) Metacloak (Liu et al., 2024). The (+) sign denotes the use of gradient ascent to construct the perturbation, whereas the (-) sign denotes the use of gradient descent, following the setting in (Xue et al., 2024). We set the perturbation budget for all evaluated protection methods to $\delta = 16/255$. Since Glaze does not release its codebase and its application does not allow adjustment of the specific perturbation budget, we use its maximum perturbation strength.

**Datasets** We evaluate the effectiveness of CAT in two main diffusion-based image synthesis applications: (1) object-driven image synthesis and (2) style mimicry. For object-driven image synthesis, we focus on human face generation using the CelebA-HQ (Karras et al., 2018) and VG-GFace2 (Cao et al., 2018) datasets. For each dataset, we select five identities, with 12 images per identity, equally split into three subsets: a clean reference subset, a target protection subset, and an additional clean reference subset. For style mimicry, we use a subset of *Claude Monet*'s artworks from the WikiArt dataset (Mancini et al., 2018), consisting of 12 images, divided into three subsets following the same configuration. All images are center-cropped and resized to $512 \times 512$.

**CAT Settings** For object-driven image synthesis, we evaluate three CAT settings: (1) CAT-both, (2) CAT-en, and (3) CAT-de. In CAT-both, CAT adapters are added to the target layers in both the encoder and decoder of the latent autoencoder. CAT-en applies CAT adapters only to the encoder, while CAT-de applies them solely to the decoder. Since we have identified that the primary reason adversarial examples are effective as protective perturbations is their latent representation distortion, the CAT-de method should have no impact on compromising the protection. We conduct this evaluation to verify our hypothesis. For style mimicry, we only evaluate (1) CAT-both, and (2) CAT-en settings.

We select all convolutional and attention layers within the module as the target layers. The adapter rank is set to $r = 128$ for the CAT-both setting and $r = 256$ for the CAT-en and CAT-de settings to maintain the same number of added parameters. During CAT, only the parameters of the attached adapters are updated, while all other weights remain frozen. Training is conducted with a batch size of 4 and a learning rate of $1 \times 10^{-4}$ for 1000 steps.

**Customization Settings** For the customization settings, we adopt the widely used DreamBooth (Ruiz et al., 2023) and LoRA (Hu et al., 2022) methods, using the latest Stable Diffusion v2.1 as the pre-trained LDM. Details of the cus-

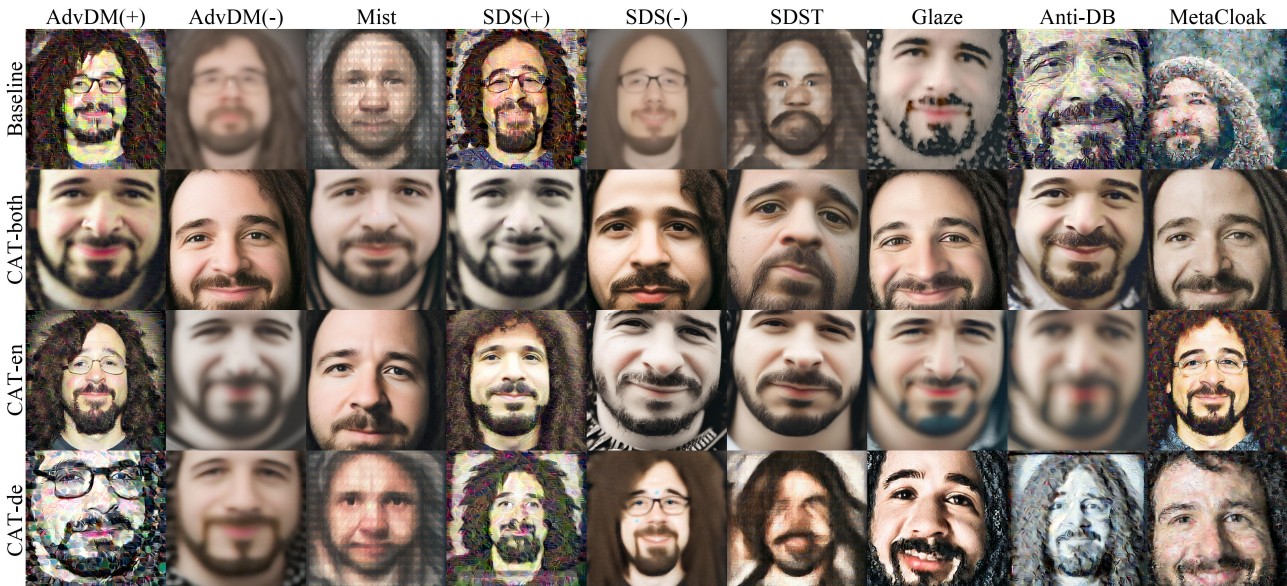

*Figure 6.* Qualitative results for object-driven image synthesis customization in an identity from the VGGFace2 dataset using DreamBooth and CAT settings with the text prompt "a dslr portrait of *sks* person". Each row represents a different setting: Baseline, CAT-both, CAT-en, and CAT-de, while each column corresponds to a different protective perturbation setting. The results illustrate the effectiveness of different CAT settings on generating human faces from protected images.

*Table 1.* Quantitative results for object-driven image synthesis using CAT methods customized in DreamBooth. The table presents performance metrics FSS and FQS for the CelebA-HQ and VGGFace2 datasets across different protective perturbation settings. The optimal values among different CAT settings for each metric are **bolded**, where higher FSS indicates improved human face similarity, and higher FQS reflects better face image quality.

| DATASET | CELEBA-HQ | | | | | | | | VGGFACE2 | | | | | | | |
|---|---|---|---|---|---|---|---|---|---|---|---|---|---|---|---|---|
| METRIC | FSS ↑ | | | | FQS ↑ | | | | FSS ↑ | | | | FQS ↑ | | | |
| METHOD | Baseline | CAT-both | CAT-en | CAT-de | Baseline | CAT-both | CAT-en | CAT-de | Baseline | CAT-both | CAT-en | CAT-de | Baseline | CAT-both | CAT-en | CAT-de |
| ADVDM(+) | 0.340 | **0.643** | 0.529 | 0.344 | 0.244 | 0.431 | **0.448** | 0.340 | 0.390 | 0.534 | **0.560** | 0.416 | 0.257 | 0.460 | **0.600** | 0.331 |
| ADVDM(-) | 0.354 | **0.623** | 0.571 | 0.401 | 0.275 | 0.549 | **0.611** | 0.401 | 0.436 | **0.564** | 0.547 | 0.454 | 0.281 | 0.617 | **0.662** | 0.492 |
| MIST | 0.160 | **0.572** | 0.501 | 0.152 | 0.286 | **0.597** | 0.580 | 0.375 | 0.035 | **0.557** | 0.521 | 0.101 | 0.384 | 0.656 | **0.712** | 0.279 |
| SDS(+) | 0.352 | **0.602** | 0.499 | 0.342 | 0.282 | 0.413 | **0.423** | 0.399 | 0.397 | 0.486 | **0.508** | 0.305 | 0.406 | 0.502 | **0.532** | 0.410 |
| SDS(-) | 0.411 | **0.678** | 0.599 | 0.405 | 0.352 | **0.597** | 0.587 | 0.449 | 0.439 | **0.570** | 0.569 | 0.435 | 0.368 | **0.730** | 0.689 | 0.456 |
| SDST | 0.175 | **0.594** | 0.485 | 0.157 | 0.208 | 0.587 | **0.588** | 0.313 | 0.146 | **0.559** | 0.546 | 0.196 | 0.248 | 0.663 | **0.694** | 0.346 |
| GLAZE | 0.363 | **0.610** | 0.577 | 0.398 | 0.491 | 0.618 | **0.676** | 0.587 | 0.387 | **0.607** | 0.576 | 0.472 | 0.576 | 0.724 | **0.740** | 0.595 |
| ANTI-DB | 0.471 | **0.662** | 0.597 | 0.489 | 0.434 | 0.608 | **0.664** | 0.546 | 0.456 | **0.584** | 0.546 | 0.440 | 0.416 | 0.673 | **0.704** | 0.401 |
| METACLOAK | 0.497 | **0.642** | 0.578 | 0.507 | 0.372 | 0.460 | **0.475** | 0.403 | 0.421 | 0.560 | **0.631** | 0.487 | 0.355 | 0.508 | **0.625** | 0.412 |

tomization settings are provided in Appendix B.

**Evaluation Metrics For Quantitative Experiments** The object-driven image synthesis task aims to generate human faces that preserve the identity features from the given protected images. Therefore, for qualitative experiments, we use Face Detection Rate (Retina-FDR), measured with the RetinaFace detector (Deng et al., 2020), to evaluate whether a face is detected in the generated images. If a face is detected, we extract its embedding using the ArcFace recognizer (Deng et al., 2019) and compute the cosine distance between the extracted embedding and those from the clean dataset, denoted as Identity Score Matching (ISM) (Le et al.,

2023). To quantify the expected identity similarity in the generated image set, we define Face Similarity Score (FSS) as the product of Retina-FDR and ISM. Moreover, we employ the TOPIQ method (Chen et al., 2024), trained on the face IQA dataset and implemented in (Chen & Mo, 2022), to measure Face Detection Rate (TOPIQ-FDR) and Face Image Quality (FIQ). To quantify the expected quality of the generated face images, we define their product as the Face Quality Score (FQS). We use two text prompts for sampling, "a photo of *sks* person" and "a dslr portrait of *sks* person". For each test prompt, we generate 30 images and compute the average metric scores across all tested identities

*Table 2.* Quantitative results for object-driven image synthesis using CAT methods customized in DreamBooth, compared to Noisy-Upscaling (NU) and Gaussian Filtering (GF) methods. The table presents performance metrics FSS and FQS for the CelebA-HQ and VGGFace2 datasets across different protective perturbation settings. The optimal values among different CAT settings for each metric are **bolded**.

| DATASET | CELEBA-HQ | | | | | | | | VGGFACE2 | | | | | | | |
|---|---|---|---|---|---|---|---|---|---|---|---|---|---|---|---|---|
| METRIC | FSS ↑ | | | | FQS ↑ | | | | FSS ↑ | | | | FQS ↑ | | | |
| METHOD | CAT-BOTH | CAT-EN | NU | GF | CAT-BOTH | CAT-EN | NU | GF | CAT-BOTH | CAT-EN | NU | GF | CAT-BOTH | CAT-EN | NU | GF |
| ADVDM(+) | **0.643** | 0.529 | 0.531 | 0.492 | 0.431 | 0.448 | **0.481** | 0.352 | 0.534 | **0.560** | 0.518 | 0.506 | 0.481 | **0.578** | 0.506 | 0.363 |
| ADVDM(-) | **0.623** | 0.571 | 0.469 | 0.607 | 0.549 | **0.611** | 0.498 | 0.526 | **0.564** | 0.547 | 0.529 | 0.563 | 0.635 | **0.676** | 0.563 | 0.506 |
| MIST | **0.572** | 0.501 | 0.491 | 0.488 | **0.597** | 0.580 | 0.475 | 0.493 | 0.557 | 0.521 | **0.566** | 0.518 | 0.662 | **0.701** | 0.518 | 0.437 |
| SDS(+) | **0.602** | 0.499 | 0.599 | 0.409 | 0.413 | 0.423 | **0.503** | 0.302 | 0.486 | **0.508** | 0.498 | 0.402 | 0.438 | **0.569** | 0.402 | 0.281 |
| SDS(-) | **0.678** | 0.599 | 0.468 | 0.583 | **0.597** | 0.587 | 0.494 | 0.493 | 0.570 | 0.569 | 0.509 | **0.593** | **0.700** | 0.671 | 0.593 | 0.558 |
| SDST | **0.594** | 0.485 | 0.470 | 0.446 | 0.587 | **0.588** | 0.474 | 0.464 | **0.559** | 0.546 | 0.521 | 0.538 | 0.627 | **0.671** | 0.538 | 0.482 |
| GLAZE | **0.610** | 0.577 | 0.533 | 0.547 | 0.618 | **0.676** | 0.496 | 0.533 | **0.607** | 0.576 | 0.503 | 0.549 | **0.733** | 0.723 | 0.549 | 0.562 |
| ANTI-DB | **0.662** | 0.597 | 0.540 | 0.575 | 0.608 | **0.664** | 0.469 | 0.543 | **0.584** | 0.546 | 0.566 | 0.548 | 0.636 | **0.656** | 0.548 | 0.499 |
| METACLOAK | **0.642** | 0.578 | 0.521 | 0.540 | 0.460 | **0.475** | 0.395 | 0.324 | 0.560 | **0.631** | 0.566 | 0.542 | 0.504 | **0.633** | 0.542 | 0.349 |

to obtain the final results. We also present evaluation results using FID (Heusel et al., 2017) and CLIP-IQA (Wang et al., 2023), which are two widely used metrics for assessing image quality.

## 5.2. Object-driven Synthesis Results

The evaluation results for applying CAT methods in the object-driven image synthesis task are presented in Table 1. We observe that both the CAT-both and CAT-en settings significantly improve task performance compared to the baseline, which directly customizes using the protected image samples across all evaluated protections. This demonstrates the effectiveness of these two settings in compromising the protective perturbations. Furthermore, the CAT-both setting achieves a higher FSS score than CAT-en, while CAT-en obtains a better FQS score overall. Additionally, the CAT-de setting has little to no effect on improving task performance. The potential reason behind this is that CAT-de only adds adapters to the decoder, which has a limited impact on realigning latent representations. Fine-tuning only the decoder is more challenging, as the diffusion model learns from distorted latents that are highly diverse, making accurate reconstruction difficult. The weaker performance of CAT-de compared to CAT-en and CAT-both further supports our observation that latent distortion is the key factor behind the effectiveness of adversarial noise as a protective perturbation. Qualitative results for one identity from the VGGFace2 dataset, as shown in Figure 6 (and for one identity from the CelebA-HQ dataset as shown in Figure 9 which is provided in Appendix C.3), further confirm the observations from the quantitative results.

Moreover, we conduct object-driven image synthesis customization experiments using LoRA as shown in Table 5, which is provided in Appendix C.1. The results verify the effectiveness of CAT in the LoRA customization setting across all nine evaluated protective perturbations and two datasets. The quantitative results measured by FID for cus-

tomization using DreamBooth are also presented in Table 6, which is provided in Appendix C.2. These results further support the effectiveness of the CAT settings in compromising the existing protective perturbation.

*Table 3.* Quantitative results for style mimicry using CAT methods customized in DreamBooth on the WikiArt dataset. The table reports CLIP-IQA scores, where higher values indicate better alignment with the target artistic style. The optimal values among different CAT settings for each metric are **bolded**.

| CLIP-IQA ↑ | BASELINE | CAT-BOTH | CAT-EN |
|---|---|---|---|
| ADVDM(+) | 0.343 | 0.390 | **0.621** |
| ADVDM(-) | 0.463 | 0.536 | **0.697** |
| MIST | 0.345 | 0.465 | **0.694** |
| SDS(+) | 0.285 | 0.366 | **0.366** |
| SDS(-) | 0.501 | 0.481 | **0.723** |
| SDST | 0.406 | 0.485 | **0.712** |
| GLAZE | 0.532 | 0.614 | **0.730** |
| ANTI-DB | 0.315 | 0.544 | **0.672** |

## 5.3. Comparison with Purification-Based Attacks

We compare our proposed CAT with two purification-based adaptive attacks against protective perturbations: Noisy-Upscaling (Hönig et al., 2025), which performs optimization-based purification, and traditional Gaussian filtering, which applies a low-pass filter to the perturbed images. Evaluations are conducted on both the CelebA-HQ and VGGFace2 datasets using CAT-both and CAT-en under the same experimental setting. For Noisy-Upscaling, we adopt the default configurations provided in the original paper. While for Gaussian filtering, we apply a Gaussian kernel of size 5 with $\sigma = 1.0$. The quantitative results are presented in Table 2. We observe that CAT consistently achieves comparable or superior performance to both Noisy-Upscaling and Gaussian Filtering across all protective perturbations, in terms of both FQS and FSS. These results highlight the competitive effectiveness of CAT compared to existing purification-based methods.

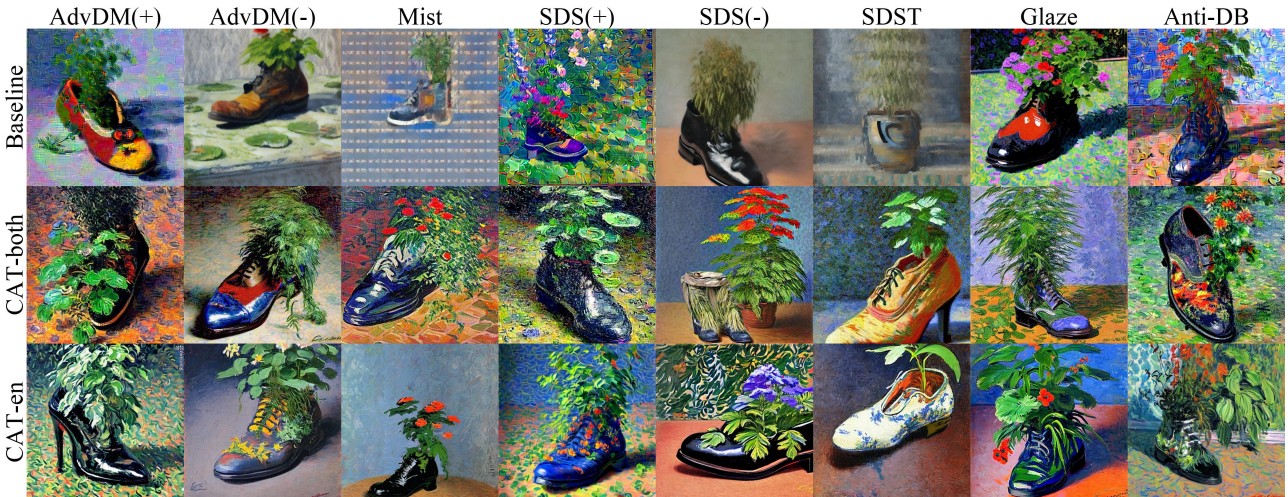

*Figure 7.* Qualitative results for style mimicry customization in DreamBooth using CAT settings with the text prompt "a painting of shoe with a plant growing inside by *claude monet* artist". Each row represents a different setting: Baseline, CAT-both and CAT-en, while each column corresponds to a different protective perturbation setting. The results illustrate the effectiveness of different CAT settings on generating artworks of the same style from protected images.

### 5.4. Style Mimicry Results

We evaluate the CAT-both and CAT-en settings for style mimicry customization using DreamBooth. Quantitative results, measured by CLIP-IQA, are presented in Table 3. CAT consistently outperforms the baseline across all evaluated protection methods, achieving higher CLIP-IQA scores and confirming the effectiveness of our approach. Qualitative comparisons are shown in Figure 7, illustrating that CAT significantly improves task performance in style mimicry customization. Across all evaluated protection perturbations, applying CAT preserves the stylistic characteristics of the target artist while improving image fidelity and mitigating the distortions introduced by protections. These results demonstrate that CAT weakens the robustness of existing protection perturbations in style mimicry, highlighting the need for more advanced protection methods.

*Table 4.* Effectiveness of the CAT-both method against AdvDM(+) protective perturbation on the VGGFace2 dataset under various CAT adapter rank settings for human face generation using DreamBooth. The optimal values for each metric are **bolded**.

| RANK | BASELINE | $r = 4$ | $r = 8$ | $r = 16$ | $r = 32$ | $r = 64$ | $r = 128$ |
|---|---|---|---|---|---|---|---|
| FSS ↑ | 0.350 | 0.525 | 0.500 | 0.513 | 0.489 | 0.540 | **0.544** |
| FQS ↑ | 0.260 | 0.372 | 0.390 | 0.458 | 0.414 | 0.381 | **0.460** |

### 5.5. Ablation Study: Impact of CAT Adapter Rank

To investigate the impact of the CAT adapter rank on the effectiveness, we conduct experiments using varying rank

settings on the VGGFace2 dataset under the AdvDM(+) protective perturbation and CAT-both setting. The results shown in Table 4 indicate that increasing the rank generally leads to better performance, with $r = 128$ achieving the highest FSS (0.544) and FQS (0.460). This demonstrates that higher-rank adapters enhance robustness against protective perturbations. However, higher-rank CAT adapters result in a larger model size. For instance, the CAT-both setting with $r = 128$ yields a 104 MB adapter model, and this size continues to increase with higher ranks. This trade-off suggests that while higher-rank adapters enhance robustness against protective perturbations, practical deployment requires balancing performance gains and model efficiency.

## 6. Conclusions

In this work, we introduce Contrastive Adversarial Training (CAT) as a novel approach to evaluating the robustness of LDM customization against protective perturbations. We identify that the primary factor contributing to the effectiveness of adversarial examples as protective perturbations lies in the distortion of their latent representations. We then propose CAT, which employs lightweight adapters trained using the contrastive adversarial loss to realign the latent representations. Extensive experiments demonstrate that CAT significantly reduces the effectiveness of a range of existing protective perturbation methods in both object-driven synthesis and style mimicry customization tasks. We believe this work can encourage the community to reconsider the robustness of existing protective perturbations and inspire the development of more resilient protections in the future.

## Acknowledgements

This work was supported by Hong Kong RGC Research Impact Fund with No.R1012-21 and General Research Fund with CityU No.11211422. We sincerely thank Linshan Hou from Harbin Institute of Technology, Shenzhen for the valuable comments and insightful suggestions.

## Impact Statement

This work evaluates the robustness limitations of existing protective perturbation methods against unauthorized data usage in diffusion-based generative models. Our goal is to highlight key vulnerabilities and provide insights toward the design of more resilient protection mechanisms. In particular, our findings suggest that effective protection may benefit from incorporating defense strategies that explicitly target the diffusion process, especially within end-to-end optimization frameworks. In contrast, relying solely on latent autoencoders may be less robust, as they are more easily compromised by adaptive attacks. This perspective points to a possible direction for developing more robust protective perturbations that better safeguard the intellectual property rights of data owners. All visual content used in this study was either publicly licensed or provided with explicit permission.

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

## A. Visualization of Latent Representations using UMAP

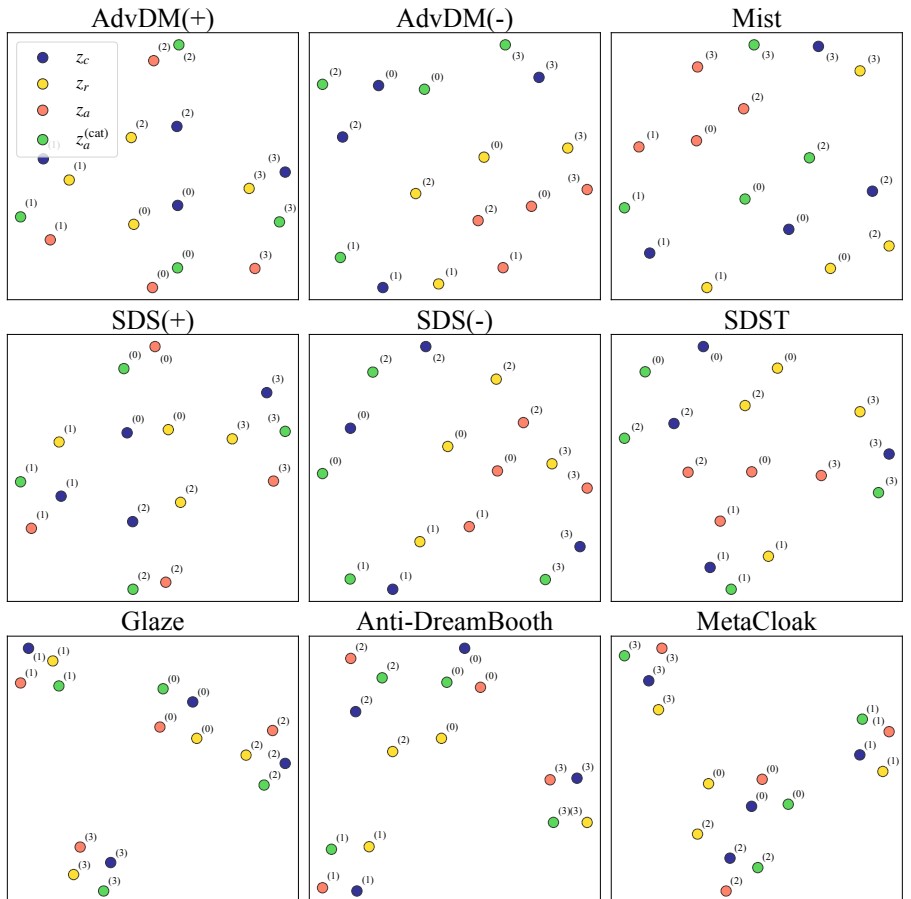

*Figure 8.* Visualization of latent representations in UMAP of clean sample $z_c$, noisy sample $z_r$, protected sample $z_a$, and the latent representation of protected sample $z_a^{(\text{cat})}$ encoded by the latent encoder with CAT adapter. Protected samples are generated using nine protective perturbation methods, with four clean images from the VGGFace2 dataset as the base. The perturbation budget for protected and noisy samples is fixed at $\delta = 16/255$. Each number labels the representations obtained from the same image.

## B. Details of Customization Settings

In the DreamBooth customization setting, both the U-Net and text encoder are fine-tuned with a batch size of 2 and a learning rate of $5 \times 10^{-7}$ for 2000 training steps. The training instance and prior class prompt for object-driven image synthesis are "a photo of *sks* person" and "a photo of person", respectively. For style mimicry, the training instance and prior class prompt are "a painting by *[ARTIST]* artist" and "a painting by artist", where *[ARTIST]* is replaced with the target artist's name. In the LoRA customization setting, LoRA adapters are added only to the U-Net with a rank of 4 and fine-tuned using a batch size of 2 and a learning rate of $5 \times 10^{-5}$ for 2000 training steps.

## C. Additional Experimental Results

### C.1. Additional Quantitative Results for Object-Driven Image Synthesis in LoRA

### C.2. Additional Quantitative Results for Object-Driven Image Synthesis in DreamBooth

### C.3. Additional Qualitative Results for Object-Driven Image Synthesis in DreamBooth

*Table 5.* Quantitative results for object-driven image synthesis using CAT methods customized in LoRA. The table presents performance metrics FSS and FQS for the CelebA-HQ and VGGFace2 datasets across different protective perturbation settings. The optimal values among different CAT settings for each metric are **bolded**.

| DATASET | CELEBA-HQ | | | | | | | | VGGFACE2 | | | | | | | |
|---|---|---|---|---|---|---|---|---|---|---|---|---|---|---|---|---|
| METRIC | FSS ↑ | | | | FQS ↑ | | | | FSS ↑ | | | | FQS ↑ | | | |
| METHOD | BASELINE | CAT-BOTH | CAT-EN | CAT-DE | BASELINE | CAT-BOTH | CAT-EN | CAT-DE | BASELINE | CAT-BOTH | CAT-EN | CAT-DE | BASELINE | CAT-BOTH | CAT-EN | CAT-DE |
| ADVDM(+) | 0.351 | **0.574** | 0.455 | 0.402 | 0.260 | 0.369 | **0.585** | 0.377 | 0.301 | **0.490** | 0.417 | 0.377 | 0.257 | 0.400 | **0.499** | 0.404 |
| ADVDM(-) | 0.424 | **0.653** | 0.608 | 0.505 | 0.235 | 0.568 | **0.686** | 0.480 | 0.367 | **0.616** | 0.598 | 0.482 | 0.281 | 0.593 | **0.729** | 0.409 |
| MIST | 0.135 | **0.565** | 0.516 | 0.005 | 0.363 | 0.469 | **0.532** | 0.022 | 0.069 | **0.596** | 0.476 | 0.001 | 0.384 | 0.565 | **0.706** | 0.000 |
| SDS(+) | 0.464 | 0.497 | **0.559** | 0.285 | 0.381 | 0.328 | **0.525** | 0.389 | 0.393 | **0.541** | 0.443 | 0.413 | 0.406 | 0.399 | **0.505** | 0.392 |
| SDS(-) | 0.476 | **0.625** | 0.586 | 0.535 | 0.297 | 0.583 | **0.665** | 0.490 | 0.353 | **0.620** | 0.591 | 0.480 | 0.368 | 0.633 | **0.695** | 0.416 |
| SDST | 0.203 | **0.572** | 0.486 | 0.032 | 0.217 | 0.488 | **0.574** | 0.137 | 0.112 | **0.549** | 0.544 | 0.052 | 0.248 | 0.578 | **0.716** | 0.217 |
| GLAZE | 0.355 | **0.669** | 0.575 | 0.479 | 0.474 | 0.714 | **0.752** | 0.617 | 0.330 | 0.602 | **0.611** | 0.472 | 0.576 | 0.673 | **0.755** | 0.586 |
| ANTI-DB | 0.539 | 0.596 | **0.616** | 0.465 | 0.497 | 0.458 | **0.541** | 0.411 | 0.460 | 0.583 | **0.586** | 0.439 | 0.416 | 0.569 | **0.725** | 0.494 |
| METACLOAK | 0.528 | **0.597** | 0.543 | 0.549 | 0.376 | 0.442 | **0.458** | 0.444 | 0.416 | **0.520** | 0.513 | 0.431 | 0.355 | 0.531 | **0.551** | 0.496 |

*Table 6.* Quantitative results for object-driven image synthesis using CAT methods customized in DreamBooth, compared to the baseline across CelebA-HQ and VGGFace2 datasets. The table reports FID scores, where lower values indicate better alignment with the real distribution. Optimal values among CAT settings are **bolded** for each dataset and method.

| FID ↓ | CELEBA-HQ | | | VGGFACE2 | | |
|---|---|---|---|---|---|---|
| METHOD | BASELINE | CAT-BOTH | CAT-EN | BASELINE | CAT-BOTH | CAT-EN |
| ADVDM(+) | 340.0 | 264.9 | **223.7** | 435.2 | 274.9 | **249.0** |
| ADVDM(-) | 134.3 | 104.0 | **102.0** | 203.9 | 189.4 | **188.6** |
| MIST | 263.6 | **133.6** | 136.1 | 359.6 | **187.8** | 198.9 |
| SDS(+) | 327.2 | 277.4 | **247.6** | 363.9 | 295.4 | **255.7** |
| SDS(-) | 125.4 | **103.4** | 110.2 | 208.9 | **183.3** | 185.6 |
| SDST | 223.0 | **133.3** | 133.5 | 335.8 | **195.3** | 200.4 |
| GLAZE | 196.7 | 100.1 | **90.4** | 228.0 | **160.9** | 191.3 |
| ANTI-DB | 180.4 | 131.4 | **106.4** | 320.5 | 202.1 | **190.8** |
| METACLOAK | 175.0 | 179.6 | **171.6** | 316.3 | 200.4 | **170.9** |

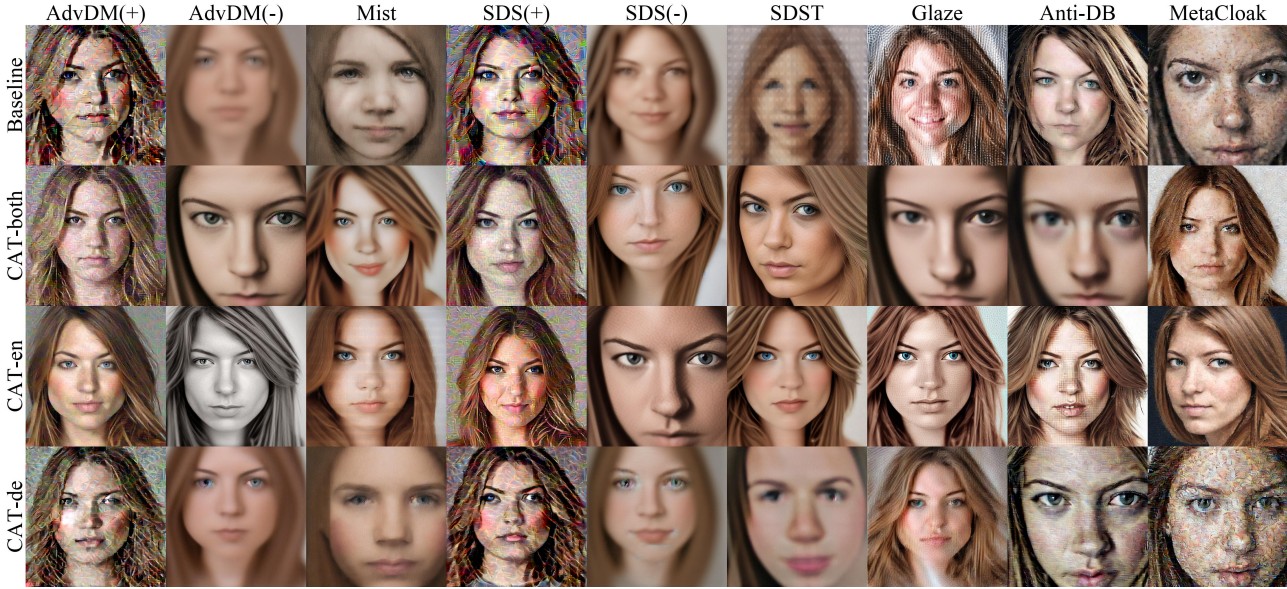

*Figure 9.* Qualitative results for object-driven image synthesis customization in an identity from the CelebA-HQ dataset using DreamBooth and CAT settings with the text prompt "a dslr portrait of *sks* person". Each row represents a different setting: Baseline, CAT-both, CAT-en, and CAT-de, while each column corresponds to a different protective perturbation setting. The results illustrate the effectiveness of different CAT settings on generating human faces from protected images.

