# OpenReview forum: "CAT: Contrastive Adversarial Training for Evaluating the Robustness of Protective Perturbations in Latent Diffusion Models"
_ICML.cc/2025/Conference — ICML 2025 poster_

### Official Review · Reviewer_jvVV · 2025-03-01

**Overall Recommendation:** 3

**Summary:**

This paper studies protective perturbations for LDMs, where the success of existing methods are based on distorted  latent representations. To examine these protections, the authors propose Contrastive Adversarial Training (CAT), which inserts lightweight adapters into the latent autoencoder. Specially, CAT realigns the latent representations, reducing the effectiveness of protective perturbations.

**Claims And Evidence:**

CAT can effectively neutralize representation distortion-based protective perturbations through contrastive adversarial training.

**Essential References Not Discussed:**

NA

**Experimental Designs Or Analyses:**

Experimental results are well organized. Authors compare CAT against nine protective perturbation methods under different customization frameworks (e.g., DreamBooth and LoRA). Both quantitative results (improvements in FSS and FQS) and qualitative examples support their claims.

**Methods And Evaluation Criteria:**

The proposed adaptive attack utilzes a contrastive adversarial loss with adapters inserted into the latent autoencoder, thereby “attacking” the distortions caused by protective perturbations.

**Other Comments Or Suggestions:**

It is good to try training-free DM customization methods (e.g., IP-Adapter).

**Other Strengths And Weaknesses:**

NA

**Questions For Authors:**

NA

**Relation To Broader Scientific Literature:**

NA

**Theoretical Claims:**

No

---

> ### Author Rebuttal · Authors · 2025-04-01
>
> We sincerely thank the reviewer for the valuable comments (Q). We hope that our responses (A) have fully addressed the concerns, and remain committed to clarifying any further questions that may arise during the discussion period.
>
> > **Q1: It is good to try training-free DM customization methods (e.g., IP-Adapter).**
>
> **A1:** We appreciate this insightful suggestion. Indeed, training-free customization methods such as IP-Adapter represent a promising direction for low-cost and scalable diffusion model customization. While our current study focuses on training-based customization approaches (e.g., DreamBooth and LoRA) due to their widespread usage and controllability in evaluating robustness, extending CAT to training-free settings is an exciting future direction. We plan to incorporate IP-Adapter into future evaluations to further evaluate the generalization of CAT beyond current training-based frameworks.
>
> **Table 3.** Quantitative results for object-driven image synthesis using CAT methods customized in DreamBooth for the CelebA-HQ dataset compared to the Noisy-Upscaling (NU) and Gaussian Filtering (GF) methods.
>
> | CelebA-HQ | FSS$\uparrow$ |        |       |       | FQS$\uparrow$ |           |           |       |
> | --------- | ------------- | ------ | ----- | ----- | ------------- | --------- | --------- | ----- |
> |           | CAT-both      | CAT-en | NU    | GF    | CAT-both      | CAT-en    | NU        | GF    |
> | AdvDM(+)  | **0.643**     | 0.529  | 0.531 | 0.492 | 0.431         | 0.448     | **0.481** | 0.352 |
> | AdvDM(-)  | **0.623**     | 0.571  | 0.469 | 0.607 | 0.549         | **0.611** | 0.498     | 0.526 |
> | Mist      | **0.572**     | 0.501  | 0.491 | 0.488 | **0.597**     | **0.580** | 0.475     | 0.493 |
> | SDS(+)    | **0.602**     | 0.499  | 0.599 | 0.409 | 0.413         | 0.423     | **0.503** | 0.302 |
> | SDS(-)    | **0.678**     | 0.599  | 0.468 | 0.583 | **0.597**     | **0.587** | 0.494     | 0.493 |
> | SDST      | **0.594**     | 0.485  | 0.470 | 0.446 | 0.587         | **0.588** | 0.474     | 0.464 |
> | Glaze     | **0.610**     | 0.577  | 0.533 | 0.547 | 0.618         | **0.676** | 0.496     | 0.533 |
> | Anti-DB   | **0.662**     | 0.597  | 0.540 | 0.575 | 0.608         | **0.664** | 0.469     | 0.543 |
> | MetaCloak | **0.642**     | 0.578  | 0.521 | 0.540 | 0.460         | **0.475** | 0.395     | 0.324 |
>
> **Table 4.** Quantitative results for object-driven image synthesis using CAT methods customized in DreamBooth for the VGGFace2 dataset compared to the Noisy-Upscaling (NU) and Gaussian Filtering (GF) methods.
>
> | VGGFace2  | FSS$\uparrow$ |           |           |           | FQS$\uparrow$ |           |       |       |
> | --------- | ------------- | --------- | --------- | --------- | ------------- | --------- | ----- | ----- |
> |           | CAT-both      | CAT-en    | NU        | GF        | CAT-both      | CAT-en    | NU    | GF    |
> | AdvDM(+)  | 0.534         | **0.560** | 0.518     | 0.506     | 0.481         | **0.578** | 0.506 | 0.363 |
> | AdvDM(-)  | **0.564**     | 0.547     | 0.529     | 0.563     | 0.635         | **0.676** | 0.563 | 0.506 |
> | Mist      | 0.557         | 0.521     | **0.566** | 0.518     | 0.662         | **0.701** | 0.518 | 0.437 |
> | SDS(+)    | 0.486         | **0.508** | 0.498     | 0.402     | 0.438         | **0.569** | 0.402 | 0.281 |
> | SDS(-)    | 0.570         | 0.569     | 0.509     | **0.593** | **0.700**     | 0.671     | 0.593 | 0.558 |
> | SDST      | **0.559**     | 0.546     | 0.521     | 0.538     | 0.627         | **0.671** | 0.538 | 0.482 |
> | Glaze     | **0.607**     | 0.576     | 0.503     | 0.549     | **0.733**     | 0.723     | 0.549 | 0.562 |
> | Anti-DB   | **0.584**     | 0.546     | 0.566     | 0.548     | 0.636         | **0.656** | 0.548 | 0.499 |
> | MetaCloak | 0.560         | **0.631** | 0.566     | 0.542     | 0.504         | **0.633** | 0.542 | 0.349 |

---

### Official Review · Reviewer_6VvR · 2025-03-13

**Overall Recommendation:** 3

**Summary:**

This paper examines the effectiveness of adversarial perturbations in protecting data from unauthorized customization in LDMs. The authors reveal that these perturbations work by distorting latent representations and propose CAT as an adaptive attack that reduces their effectiveness. Experimental results highlight the vulnerability of current protection methods.

**Claims And Evidence:**

Most claims are supported by clear evidence.

**Essential References Not Discussed:**

No

**Experimental Designs Or Analyses:**

The study is reasonably demonstrated, but some expectations are not met. There are no quantitative results for style mimicry and in addition, there is no comparisons with other purification methods. The authors only provide results with and without their method, which makes the effectiveness of the proposed approach unconvincing.

**Methods And Evaluation Criteria:**

Yes, but could be improved. For instance, in Table 1 or for the style mimicry scenario, it would be more insightful to evaluate using the fidelity metrics, such as FID.

**Other Comments Or Suggestions:**

No.

**Other Strengths And Weaknesses:**

The core rationale behind the proposed method is reasonable. However, as noted in the “Experimental Design” section, the lack of sufficient experiments and analysis limits the convincingness of the work.

**Questions For Authors:**

I am curious about the recent trend of research primarily focusing on mimicry attacks to circumvent existing protection methods. I think this approach inherently has limitations in advancing genuinely robust protection methods, including the current study. Given this purification, how might we move towards stronger protection methods, or at least derive valuable insights to guide future research directions? It would be very insightful and helpful to the future reader.

**Relation To Broader Scientific Literature:**

The intuition and experiments focus on mimicry tasks, which have emerged as significant concerns in the context of generative AI, particularly regarding copyright issues.

**Theoretical Claims:**

No theoretical claims.

---

> ### Author Rebuttal · Authors · 2025-04-01
>
> We sincerely thank the reviewer for the valuable comments (Q). We hope that our responses (A) have fully addressed the concerns, and remain committed to clarifying any further questions that may arise during the discussion period.
>
> > **Q1: Yes, but could be ... using the fidelity metrics, such as FID.**
>
> **A1:** We thank the reviewer for pointing this out. In the evaluation of **Table. 1 below**, we use the FID metric to assess generation quality. Specifically, for each identity, we generate 30 images per prompt (two prompts in total), and compute FID by comparing the results to images generated by a customization model trained on clean samples. We observe that both CAT-both and CAT-en consistently achieve lower FID scores than the baseline across both datasets and all evaluated protections. These results demonstrate the effectiveness of our CAT under the evaluated settings.
>
> **Table 1.** Quantitative results for object-driven image synthesis using CAT methods customized in DreamBooth for CelebA-HQ and VGGFace2 datasets compared to the Noisy-Upscaling (NU) and Gaussian Filtering (GF) methods.
>
> | FID$\downarrow$ | CelebA-HQ |           |           | VGGFace2 |           |           |
> | --------------- | --------- | --------- | --------- | -------- | --------- | --------- |
> |                 | Baseline  | CAT-both  | CAT-en    | Baseline | CAT-both  | CAT-en    |
> | AdvDM(+)        | 340.0     | 264.9     | **223.7** | 435.2    | 274.9     | **249.0** |
> | AdvDM(-)        | 134.3     | 104.0     | **102.0** | 203.9    | 189.4     | **188.6** |
> | Mist            | 263.6     | **133.6** | 136.1     | 359.6    | **187.8** | 198.9     |
> | SDS(+)          | 327.2     | 277.4     | **247.6** | 363.9    | 295.4     | **255.7** |
> | SDS(-)          | 125.4     | **103.4** | 110.2     | 208.9    | **183.3** | 185.6     |
> | SDST            | 223.0     | **133.3** | 133.5     | 335.8    | **195.3** | 200.4     |
> | Glaze           | 196.7     | 100.1     | **90.4**  | 228.0    | **160.9** | 191.3     |
> | Anti-DB         | 180.4     | 131.4     | **106.4** | 320.5    | 202.1     | **190.8** |
> | MetaCloak       | 175.0     | 179.6     | **171.6** | 316.3    | 200.4     | **170.9** |
>
>
> > **Q2: The core rationale ... proposed approach unconvincing.**
>
> **A2:** We sincerely thank the reviewer for the valuable suggestion. To the best of our knowledge, this is the first work that systematically evaluates the robustness of nine existing protective perturbation methods across two downstream tasks, which requires careful data preparation and repeated experiments. That said, we totally agree that for the style mimicry task, only qualitative results were provided in the main paper due to space limitations. To address this, we include quantitative results **in Table 2 below** in CLIP-IQA score [r2] for both CAT-both and CAT-en settings. We observe that in the style mimicry task, CAT consistently outperforms the baseline across all evaluated protection methods, further demonstrating the effectiveness of our approach.
>
> **Table 2.** Quantitative results for style mimicry using CAT methods customized in DreamBooth for the WikiArt dataset.
>
> | CLIP-IQA$\uparrow$ | Baseline | CAT-both | CAT-en    |
> | ------------------ | -------- | -------- | --------- |
> | AdvDM(+)           | 0.343    | 0.390    | **0.621** |
> | AdvDM(-)           | 0.463    | 0.536    | **0.697** |
> | Mist               | 0.345    | 0.465    | **0.694** |
> | SDS(+)             | 0.285    | 0.366    | **0.366** |
> | SDS(-)             | 0.501    | 0.481    | **0.723** |
> | SDST               | 0.406    | 0.485    | **0.712** |
> | Glaze              | 0.532    | 0.614    | **0.730** |
> | Anti-DB            | 0.315    | 0.544    | **0.672** |
>
> > **Q3: I am curious about the ... and helpful to the future reader.**
>
> **A3:** Thank you for the insightful comment. We fully agree that while adaptive attacks reveal current vulnerabilities in protective perturbations, their broader value lies in informing the development of more effective defenses. In this regard, we would like to emphasize that our CAT method not only exposes the limitations of existing protections but also offers potential insights for future defense strategies. In particular, our findings suggest that effective protection for LDMs may benefit from incorporating defense mechanisms that explicitly consider the diffusion process, especially in end-to-end optimized frameworks, rather than relying solely on latent autoencoders, which can be more easily compromised by adaptive attacks. We believe this perspective highlights a promising direction for designing more robust protective perturbations to better safeguard the IP rights of data owners. A more detailed discussion will be added in the revised version.
>
> [r2] Wang, J., Chan, K. C., & Loy, C. C. (2023, June). Exploring clip for assessing the look and feel of images. In *Proceedings of the AAAI conference on artificial intelligence* (Vol. 37, No. 2, pp. 2555-2563).

---

### Official Review · Reviewer_M2AM · 2025-03-17

**Overall Recommendation:** 3

**Summary:**

This paper investigates adversarial examples as protective perturbations in latent diffusion models. The authors reveal that the reason why adversarial examples are effective is primarily due to the distortion of their latent representations. Based on this observation, they propose the CAT method to attack protective methods, highlighting their lack of robustness.

**Claims And Evidence:**

The claim that the experiments in section 3 explain why adversarial examples are effective as protective perturbations seems an overclaim to me. The most direct conclusions of the two experiments in section 3 are that adversarially perturbed images lead to larger distortions in latent representations and that the diffusion model is able to learn adversarial examples. It is expected to have an explicit measure of generation degradation to show the correlation between large distortions and protection effectiveness. Even if there is a strong correlation, the causality should be carefully claimed because there may be other potential factors.

**Essential References Not Discussed:**

None

**Experimental Designs Or Analyses:**

See “Claims and Evidence” part. Other experiments are sound.

**Methods And Evaluation Criteria:**

The proposed method and evaluation  makes sense for the task.

**Other Comments Or Suggestions:**

All the quotation marks should be “ ” instead of ” ”.

The header shows “Submission and Formatting Instructions for ICML 2024”.

**Other Strengths And Weaknesses:**

Strengths:

A deeper investigation into how adversarial examples are effective as protective perturbations helps better understand this kind of method.

The model-based adaptive attack method is different from previous purification-based attacks, which shows the novelty of this work.

The authors evaluate the attack against nine protection methods, demonstrating the effectiveness of the proposed attack.

Weaknesses:

See “Claims and Evidence” part. The conclusion in section 3 seems an overclaim to me. If I misunderstood, please correct me.

Lack of baseline methods. Even if the model-based adaptive attack is novel, I still suggest comparing it with previous methods (e.g. IMPRESS++[1]) to show whether the model-based adaptive attack is more effective than other methods.

The evaluation metrics “FSS” and “FQS” are not standard but reasonable. Separate reports of Retina-FDR and ISM may enhance understanding of the results. And so do TOPIQ-FDR and FIQ. In addition, some traditional evaluation metrics like PSNR and SSIM are also expected to be adopted to show the quality of the generated images.

[1] Hönig, Robert, et al. "Adversarial perturbations cannot reliably protect artists from generative ai." arXiv preprint arXiv:2406.12027 (2024).

**Questions For Authors:**

According to Table 1, the “CAT-de” is consistently weaker than “CAT-both” and “CAT-en”, and is sometimes even weaker than baseline. What are the potential reasons behind this?

Since “CAT-both” and “CAT-en” are comparable on different metrics, how to choose which method to use in practice?

**Relation To Broader Scientific Literature:**

Latent Representation Distortion: Previous works [1,2] built adversarial examples to protect diffusion models. However, a deeper investigation into how the adversarial examples work is lacking. This paper conducts qualitative and quantitative experiments to analyze adversarial examples as protective methods.

Adaptive Attack: Most existing attacks rely on purification. This paper proposes a model-based adaptation method.

[1] Liang, Chumeng, et al. "Adversarial example does good: Preventing painting imitation from diffusion models via adversarial examples." arXiv preprint arXiv:2302.04578 (2023).

[2] Xue, Haotian, et al. "Toward effective protection against diffusion-based mimicry through score distillation." The Twelfth International Conference on Learning Representations. 2023.

[3] Hönig, Robert, et al. "Adversarial perturbations cannot reliably protect artists from generative ai." arXiv preprint arXiv:2406.12027 (2024).

**Theoretical Claims:**

no theoretical claims in this paper

---

> ### Author Rebuttal · Authors · 2025-04-01
>
> We sincerely thank the reviewer for the valuable comments (Q). We will correct the identified typos and incorporate the suggested references you mentioned. We hope that our responses (A) have fully addressed the concerns, and remain committed to clarifying any further questions that may arise during the discussion period.
>
>
> > **Q1: The claim ... other potential factors.**
>
> **A1:** We would like to clarify that the observed strong correlation refers to the effectiveness of adversarial noise as protective perturbations and the distortion in their latent representations. This conclusion is supported by the following observations:
>
> 1. **Latent representation distortion:** As shown in Fig. 3, adversarial noise (red dots) causes significantly more distortion in the latent space compared to random perturbations (yellow dots) under the same budget. After applying CAT, the adversarial samples (green dots) are noticeably re-aligned with the clean samples (blue dots). This observation is quantitatively verified in Fig. 4.
> 2. **Learnability of perturbed latent representations:** Fig. 5 shows that adversarially perturbed latent representations can still be effectively learned by the diffusion model, with learnability comparable to clean and randomly perturbed samples.
> 3. **Degradation on generation quality:** Fig. 6 and Table 1 in the manuscript present results when fine-tuning on adversarially perturbed data. Without CAT (baseline), generation quality drops significantly, while applying CAT improves quality to a large extent.
>
> Together, these obervations and experimental results support our conclusion that the unlearnability of adversarially perturbed samples primarily comes from their latent representation distortion. That said, we fully agree with the reviewer that, even in the presence of a strong correlation, causality should be claimed with caution, as other factors may also play a role. We thank the reviewer for this important insight, and will emphasize that this conclusion is mainly based on empirical experimental observations
>  and there are other potential factors.
>
>
> > **Q2: Lack of baseline ... methods.**
>
> **A2:** We have compared our proposed CAT with baseline adaptive attacks against protective perturbations: Noisy-Upscaling [r1] (optimization-based) and Gaussian Filtering (low-pass filtering-based). *IMPRESS++ is not open-sourced yet, so we instead adopt Noisy-Upscaling, the superior adaptive attack from the same paper.* **Due to space limitations, experimental details are provided in our response to Reviewer GQca, with results on the CelebA-HQ dataset shown in Table 3, and on the VGGFace2 dataset in Table 4 (both in response to Reviewer jvVV).** We apologize for any inconvenience this may cause and appreciate your understanding. We can observe that our proposed CAT (CAT-both and CAT-en) consistently achieves comparable or superior performance to both Noisy-Upscaling and Gaussian Filtering across all protective perturbations, in terms of both FQS and FSS. These results highlight the competitive effectiveness of CAT compared to existing purification-based methods.
>
>
> > **Q3: The evaluation ... generated images.**
>
> **A3:** We appreciate the reviewer for bringing this to our attention. We will report the results for Retina-FDR and ISM separately, as well as those for TOPIQ-FDR and FIQ, in the supplementary materials. **Due to space limitations, we present the traditional evaluation metric FID to assess the quality of generated images in Table 1 (in response to Reviewer 6VvR).** It can be observed that our proposed CAT (either CAT-both or CAT-en) consistently improves generation quality in terms of FID across both datasets and all evaluated protection methods. These results demonstrate the effectiveness of our approach.
>
> > **Q4: According to Table 1 ... behind this?**
>
> **A4:** The potential reason behind this that CAT-de only adds adapters to the VAE decoder, which has limited impact on realigning latent representations. Fine-tuning only the decoder is more challenging, as the diffusion model learns from distorted latents that are highly diverse, making accurate reconstruction difficult. The weaker performance of CAT-de compared to CAT-en and CAT-both further supports our observation that latent distortion is the key factor behind the effectiveness of adversarial noise as a protective perturbation.
>
>
> > **Q5: Since “CAT-both” ... use in practice?**
>
> **A5:** In our experiments, we keep the parameter size the same for CAT-both and CAT-en. Specifically, CAT-both uses half the adapter rank of CAT-en, as it adds adapters to both the encoder and decoder. While their performance varies across tasks, datasets, and metrics, the overall results are comparable. That said, we speculate that CAT-en may perform better on tasks requiring more specific or dense latent representations, **such as the style mimicry task in Table 2 (in response to Reviewer 6VvR)**, as it enables stronger latent alignment during customization.

---

### Official Review · Reviewer_GQca · 2025-03-18

**Overall Recommendation:** 3

**Summary:**

This paper proposes an attack named CAT that can break the protection of preventing diffusion models from effectively learning unauthorized data being perturbed by defensive noise. The authors first empirically identify that the mechanism behind existing defensive perturbations is to make embeddings of perturbed images look different from the embeddings of clean images. Based on this observation, the authors propose to break the protection via improving the robustness of the encoder in diffusion models against those defensive perturbations. Experiments are conducted on the stable-diffusion-v2.1 model with various protection methods.

## **update after rebuttal**

After reading the rebuttal, I think this paper has novel results. So I decide to maintain my current score but tend to acceptance.

For the authors, they should update the paper to include:

- Additional results on data-augmentations.

- Discussions on RobustCLIP.

**Claims And Evidence:**

See **Weaknesses/Questions/Suggestions**.

**Essential References Not Discussed:**

The concept of "preventing unauthorized image usage via defensive noises" originally comes from the research of "unlearnable examples". Therefore, the authors should also cite and review papers on "unlearnable examples" such as:

1. Huang et al. "Unlearnable Examples: Making Personal Data Unexploitable." ICLR 2021.

2. Fu et al. "Robust Unlearnable Examples: Protecting Data Against Adversarial Learning." ICLR 2022.

3. Ren et al. "Transferable Unlearnable Examples." ICLR 2023.

4. Liu et al. "Stable unlearnable example: Enhancing the robustness of unlearnable examples via stable error-minimizing noise." AAAI 2024.

**Experimental Designs Or Analyses:**

See **Weaknesses/Questions/Suggestions**.

**Methods And Evaluation Criteria:**

See **Weaknesses/Questions/Suggestions**.

**Other Comments Or Suggestions:**

See **Weaknesses/Questions/Suggestions**.

**Other Strengths And Weaknesses:**

**Strengths:**

1. I like the observation of identifying the mechanism behind existing protective perturbations. It is intuitively sound and makes sense.


**Weaknesses/Questions/Suggestions:**

1. I think the authors should compare their proposed method with some simple image augmentation-based defenses, for example, various low-pass filters.

2. Two major drawbacks of the proposed method are that to perform adversarial training following Eq.(1), the adversary needs to: (1) know in advance what defensive noise is leveraged by $x_a$, and (2)  generate a set of protected training images for the specified defensive noise.

3. I think the AT method from RobustCLIP [r1], which aims to enhance the adversarial robustness of CLIP image encoder against input perturbation within a pre-defined radius, is much better than the proposed CAT method. Although this method is originally designed for CLIP, it can be directly applied to any image encoders such as the VAE used in stable-diffusion-v2.1. Under the RobustCLIP's AT method, the adversary can train the VAE encoder only once to make it be robust to multiple types of protective perturbations. However, the proposed CAT method needs to retrain the VAE encoder every time for new defensive noise.

4. I suggest the authors add a small section explaining how the stable diffusion works with encoders. The current paper is very difficult for readers without any background knowledge about stable diffusion to understand the threat model of this work.

5. What do those blue points $z_0$ mean in Fig.3? It seems that they are never explained in the paper.


**Reference**

[r1] Schlarmann et al. "Robust CLIP: Unsupervised Adversarial Fine-Tuning of Vision Embeddings for Robust Large Vision-Language Models." ICML 2024.

**Questions For Authors:**

See **Weaknesses/Questions/Suggestions**.

**Relation To Broader Scientific Literature:**

N/A

**Theoretical Claims:**

N/A

---

> ### Author Rebuttal · Authors · 2025-04-01
>
> We sincerely thank the reviewer for the valuable comments (Q). We will incorporate the suggested references you mentioned. We hope that our responses (A) have fully addressed the concerns, and remain committed to clarifying any further questions that may arise during the discussion period.
>
> > **Q1: I think the authors ... various low-pass filters.**
>
> **A1:** We have compared our proposed CAT with two image augmentation-based adaptive attacks towards the protective perturbation: Noisy-Upscaling [r1] (based on optimization) and Gaussian filtering (based on low-pass filters). The evaluation is conducted on both the CelebA-HQ and VGGFace2 datasets using CAT-both and CAT-en, following the same experimental setting. For Noisy-Upscaling, we adopt default configurations provided in the original paper, and for Gaussian Filtering, we use a Gaussian kernel with size equal to 5 and $\sigma = 1.0$. **Due to space limitations, we present the results on the CelebA-HQ  in Table 3 and on the VGGFace2 dataset in Table 4 (both in response to Reviewer jvVV)**. We apologize for any inconvenience this may cause and appreciate your understanding. We can observe that our proposed CAT (CAT-both or CAT-en) consistently achieves comparable or superior performance to both Noisy-Upscaling and Gaussian Filtering across all protective perturbations, in terms of both FQS and FSS. These results highlight the competitive effectiveness of CAT compared to existing purification-based methods.
>
> > **Q2: Two major drawbacks... for the specified defensive noise.**
>
> **A2:** We would like to clarify two key aspects of our threat model: the data owner has the clean data $x_c$ and intends to share it publicly. To prevent unauthorized customization, the owner applies a perturbation to generate the protected data $x_a$, which is then released. **In this setting, the adversary only has access to the protected data $x_a$ and is unaware of the specific protection method used by the data owner. This protected data already contains the defensive noise, and no additional generation is required by the adversary.** We want to demonstrate that, even without knowledge of the protection technique and with access only to the protected data, the adversary can still effectively learn using our proposed CAT. This is achieved by using our proposed **contrastive adversarial loss** during customization. We will include a more detailed explanation of the threat model in the revised version.
>
> > **Q3: I think the AT method from RobustCLIP [r1] ... for new defensive noise.**
>
> **A3:** We fully agree that RobustCLIP presents an effective adversarial training (AT) framework for enhancing the robustness of CLIP-based vision encoders, particularly against perturbations that disrupt text-image semantic alignment. That said, we would like to clarify key differences between RobustCLIP and our proposed CAT:
>
> 1. **Optimization objective:** RobustCLIP aims to *preserve semantic alignment between text and image representations*. In contrast, CAT defends against protective perturbations that distort the latent representation in the autoencoder, using a *contrastive adversarial loss* to explicitly enforce latent space alignment.
> 2. **Implementation framework:** While it is possible to apply RobustCLIP to the VAE encoder in LDMs, this would involve fine-tuning the VAE on self-generated adversarial-clean pairs, which is the same idea aligned with CAT’s motivation. However, *CAT achieves this via lightweight adapters* applied during customization on protected data. These adapters are detachable at inference time, leaving the original model performance unaffected. In contrast, fine-tuning of RobustCLIP would require significantly more data and adversarial examples, and would modify the entire encoder, potentially degrading performance on the original task.
>
> Despite these differences, we greatly appreciate the reviewer’s suggestion and will discuss it in the related work section to highlight these differences.
>
>
> > **Q4: I suggest the authors add a small section ... of this work.**
>
> **A4:** We totally agree that a brief explanation of how Stable Diffusion operates as an LDM with the latent autoencoder would improve the paper’s readability for those unfamiliar with this background. We will add a dedicated section to clarify this in the related work.
>
> > **Q5: What do those blue points $z_0$ mean in Fig.3? It seems that they are never explained in the paper.**
>
> **A5:** We apologize for the confusion caused by the incorrect labeling in the original Fig. 3. The blue points were intended to represent $z_c$， which is the latent embeddings of clean samples as correctly noted in the caption. We have updated both the figure and its caption to reflect this correction. Thank you for pointing this out.
>
> [r1] Hönig, R., Rando, J., Carlini, N., & Tramèr, F. Adversarial Perturbations Cannot Reliably Protect Artists From Generative AI. In *The Thirteenth International Conference on Learning Representations*.

---

> > ### Comment · Reviewer_GQca · 2025-04-04
> >
> > Thanks to the authors for their rebuttal, which resolved most of my concerns. Therefore, I will maintain my current score but tend to accept this paper.
> >
> > Please include: (1) additional results on data-augmentations, and (2) discussions on RobustCLIP in your revised paper.

---

> > > ### Author Response · Authors · 2025-04-08
> > >
> > > Thank you for your thoughtful and positive feedback. We're glad to hear that most of your concerns have been resolved, and we appreciate your support. As you suggested, we will make sure to include additional experiments and a discussion on RobustCLIP in the revised version of the paper. Thank you again for your helpful comments.

---

### Decision · Program_Chairs · 2025-05-01

**Decision:**

Accept (poster)

**Comment:**

This paper was reviewed by four experts in the field and finally received overall borderline scores: Weak accept, Weak accept, Weak accept, and Weak accept.
The major concerns of the reviewers are:
1.	 inconsistencies in method design and motivation,
2.	additional results on data-augmentations, other metrics, and comparisons with other methods,
3.	insufficient experiments results analysis.
The authors address most of the above concerns during the discussion period. Hence, I make the decision to accept the paper if there is room in the program. However, if the paper is finally accepted, the authors should carefully revise it according to the reviewers’ comments.